# Impact of Pre-Transplant Left Ventricular Diastolic Pressure on Primary Graft Dysfunction after Lung Transplantation: A Narrative Review

**DOI:** 10.3390/diagnostics14131340

**Published:** 2024-06-24

**Authors:** Jean Philippe Henry, François Carlier, Julien Higny, Martin Benoit, Olivier Xhaët, Dominique Blommaert, Alin-Mihail Telbis, Benoit Robaye, Laurence Gabriel, Antoine Guedes, Isabelle Michaux, Fabian Demeure, Maria-Luiza Luchian

**Affiliations:** 1Department of Cardiology, Université Catholique de Louvain, CHU UCL Namur, 5530 Yvoir, Belgium; julien.higny@chuuclnamur.uclouvain.be (J.H.); martin.benoit@chuuclnamur.uclouvain.be (M.B.); olivier.xhaet@chuuclnamur.uclouvain.be (O.X.); dominique.blommaert@chuuclnamur.uclouvain.be (D.B.); mihail_telbis@yahoo.co.uk (A.-M.T.); benoit.robaye@chuuclnamur.uclouvain.be (B.R.); laurence.gabriel@chuuclnamur.uclouvain.be (L.G.); antoine.guedes@chuuclnamur.uclouvain.be (A.G.); fabian.demeure@chuuclnamur.uclouvain.be (F.D.); marialuiza.luchian@yahoo.com (M.-L.L.); 2Department of Pneumology, Université Catholique de Louvain, CHU UCL Namur, 5530 Yvoir, Belgium; francois.carlier@chuuclnamur.uclouvain.be; 3Department of Intensive Care, Université Catholique de Louvain, CHU UCL Namur, 5530 Yvoir, Belgium; isabelle.michaux@chuuclnamur.uclouvain.be

**Keywords:** lung transplantation, diastolic dysfunction, primary graft dysfunction, left ventricle function, E/e’

## Abstract

Lung transplantation (LT) constitutes the last therapeutic option for selected patients with end-stage respiratory disease. Primary graft dysfunction (PGD) is a form of severe lung injury, occurring in the first 72 h following LT and constitutes the most common cause of early death after LT. The presence of pulmonary hypertension (PH) has been reported to favor PGD development, with a negative impact on patients’ outcomes while complicating medical management. Although several studies have suggested a potential association between pre-LT left ventricular diastolic dysfunction (LVDD) and PGD occurrence, the underlying mechanisms of such an association remain elusive. Importantly, the heterogeneity of the study protocols and the various inclusion criteria used to define the diastolic dysfunction in those patients prevents solid conclusions from being drawn. In this review, we aim at summarizing PGD mechanisms, risk factors, and diagnostic criteria, with a further focus on the interplay between LVDD and PGD development. Finally, we explore the predictive value of several diastolic dysfunction diagnostic parameters to predict PGD occurrence and severity.

## 1. Introduction

Lung transplantation (LT) currently constitutes the ultimate standard therapy in highly selected patients with end-stage lung diseases (ESLDs). Although transplantation techniques, organ preservation, and medical management following LT have steadily improved during the past 30 years, primary graft dysfunction (PGD) remains a frequent issue in LT recipients [1]. PGD constitutes a form of severe lung injury, occurring in the first 72 h after LT, which has been named ischemia-reperfusion injury, reperfusion edema, pulmonary reimplantation response, or early allograft dysfunction, underlying its intricate driving mechanisms. The pathogenesis of PGD is complex and the primary causative mechanism is consequent to ischemia-reperfusion injury (IRI), resulting from direct damage of ischemia and preservation, the generation of reactive oxygen species (ROS) at reperfusion, and activation of a damage-amplifying proinflammatory cascade [2]. 

Some studies suggest that graft reperfusion, rather than ischemia itself, may cause injury, further increasing the risk of PGD onset [2,3]. The activation and release of inflammatory mediators and signals from the donor lung can intensify post-transplant inflammation [2,3]. Nevertheless, these inflammatory signals do not always result in the clinical symptoms of PGD, as post-transplant outcomes depend on factors related to both the donor lung and the recipient [3]. Therefore, a comprehensive assessment of the recipient’s diastolic function may potentially influence the clinical progression of LT and PGD occurrence in certain patients. Additionally, previous reports indicated that donor characteristics can influence LT outcomes especially in the first 24 h, while recipient-related factors become more significant in the days following LT [3].

Currently, PGD is a clinical syndrome defined by hypoxemia and alveolar infiltrates on chest X-rays, which constitutes the most common cause of early mortality following LT [1,4,5]. Importantly, PGD also constitutes an important risk factor to develop chronic lung allograft dysfunction, which remains the first cause of death in LT recipients beyond the first year [6,7,8]. In LT candidates, a secondary enhancing causative mechanism of PGD has appeared. The presence of pulmonary hypertension (PH) recently emerged as an important risk factor for developing PGD [4,9]. While pre-LT PH is usually due to respiratory status (group 3 PH) [10], the role of left ventricular diastolic dysfunction (LVDD) in promoting PH (and eventually PGD) in LT candidates remains largely overlooked. LVDD is characterized by a decreased myocardial relaxation and increased left ventricle end-diastolic pressure (LVEDP) [11] and is thought to also influence the clinical course following LT [12,13]. 

Currently, data exploring the mechanisms underlying the interplay between LVDD and PGD onset remain scarce. This review summarizes the current knowledge on this presumed association, while emphasizing the role of multimodality imaging in evaluating subclinical cardiac dysfunction in LT candidates.

An extensive literature review was conducted, recapitulating research studies involving exclusively human subjects, published in English, and indexed in MEDLINE (through PubMed). The research keywords included but were not limited to the following: lung transplantation, diastolic dysfunction, primary graft dysfunction, pulmonary hypertension, right ventricle, mitral pattern, left atrial volume, E over e’, and chronic lung allograft dysfunction. 

## 2. Primary Graft Dysfunction—Role of Pulmonary Hypertension 

Since its emergence in the early 1980s, LT has steadily become the standard therapy for patients suffering from ESLDs, such as interstitial lung diseases, chronic obstructive pulmonary disease (COPD), pulmonary arterial hypertension (PAH), cystic fibrosis, and other less common diseases. Although significant progress has been made over the past 30 years, both short- and long-term survival after LT remains low in comparison to other solid-organ transplants, with a 1-year mortality of 15% and a median survival of approximately 6.5 years [8]. 

PGD of a transplanted lung is the most common cause of early mortality, accounting for 20% of deaths occurring in the first 30 days after LT [1]. In the context of hypoxemia and ruling out alternative clinical causes, diffuse pulmonary opacities upon thoracic imaging characterize this potentially fatal LT complication [12].

By definition, it occurs within the first 72 h following transplantation and encompasses a range of lung damage, varying in intensity from mild to severe, determined by the level of hypoxemia and extent of lung injury [13].

Risk factors for developing PGD that have been described range from donor characteristics and surgical protocols to factors related to the LT recipient (Table 1) [14]. The mechanisms underlying PGD are complex and incompletely understood. Ischemia and reperfusion in allografts initiate a cascade of interrelated cellular processes, leading to extensive damage to the vascular endothelium, alveolar epithelial cells death, and inflammatory cytokine/chemokine release [15,16]. This process, which is the primary causative mechanism, is further characterized by oxidative stress and innate immune system activation [15,16]. Consequently, pulmonary capillary permeability and pulmonary vascular resistance both increase, while gas exchange and pulmonary compliance decrease [16]. The exacerbation of these mechanisms significantly contributes to developing chronic lung allograft dysfunction and bronchiolitis obliterans syndrome (BOS), considerably impeding functional outcomes [13,15]. In LT recipients, factors such as female gender, body mass index (BMI) > 25 kg/m^2^, and elevated pulmonary artery systolic pressure (sPAP) have been linked to PGD onset and validated in prospective prediction models [17,18,19]. A meta-analysis, including 13 studies and encompassing over 10,000 LT recipients, highlighted the significance of primary LT candidate care assessments. It confirmed that pre-existing PH and elevated sPAP are closely linked to PGD incidence [9]. In line with these observations, a recent study including 158 consecutive LT patients proposed a cut-off of 37 mmHg for post-reperfusion sPAP as an independent PGD predictor [20], being superior to pre-reperfusion sPAP to predict PGD. 

Nevertheless, reliance should not be placed only on specific mean pulmonary artery pressure thresholds when aiming at predicting PGD. Indeed, the broad spectrum of cardiac alterations linked to PH in LT recipients needs to be considered. Additionally, a range of factors related to right ventricle function (RVF) may also play a key role in the onset of PGD. While a certain degree of right ventricular dysfunction is typically regarded as normal in LT candidates, some exhibit a compensated or ‘pseudonormal’ RVF, often due to a hypertrophic right ventricle (RV), which is a frequent occurrence in PH patients [21]. 

RVF is a dynamic, rapidly evolving factor in the progression of LT. Kusunose et al. retrospectively assessed 89 patients, comparing RVF pre- and post-LT, and found notable decreases in RV areas (*p* = 0.033 and *p* < 0.001), sPAP (*p* < 0.001), and pulmonary vascular resistance (PVR) (*p* < 0.001) following LT [22]. Improvements were also seen in the RV fractional area change (*p* < 0.001) and RV strain (*p* = 0.021) [22]. Additionally, a study including 72 LT patients observed that those showing enhanced RVF, as determined by speckle-tracking imaging, were more susceptible to develop severe PGD, due to increased pulmonary flow [21]. However, a limitation of this study was its inadequate examination of LVDD, which can both affect RVF and subsequently impact early postoperative outcomes [21]. These findings parallel the observations of Kusunose et al., who also described significant changes in RV parameters post-LT. The circulatory system being a closed loop, the interaction of both ventricles cannot be neglected. Alterations in one ventricle’s hemodynamic affects the other, and RV dysfunction has been shown to cause a diminution of the left ventricle (LV) performance, irrespectively of its mechanisms [23]. Therefore, both ventricles play a crucial role in the onset of PGD and should be considered in its management. However, the role of LVDD in PGD following LT remains largely overlooked, and research in the field started only a few years ago. 

## 3. Primary Graft Dysfunction—Left Ventricular Diastolic Dysfunction Current Guidelines’ Evaluation

LVDD is associated with a decreased myocardial relaxation and an increased LVEDP, elevating left atrial pressure (LAP) and causing pulmonary venous congestion. As per 2016 guidelines, diastolic dysfunction is identified by four echocardiographic variables: septal e’ < 7 cm/s, lateral e’ < 10 cm/s, average E/e’ ratio > 14, left atrium (LA) volume index > 34 mL/m^2^, and tricuspid regurgitation velocity > 2.8 m/s [24]. Two algorithms, based on normal or abnormal left ventricle ejection fraction (LVEF), utilize these variables. LVDD classification requires at least three of four variables to meet the criteria. Definite LVDD needs more than half of the parameters to meet the cut-off values, whereas 0–1 relates to normal left ventricle diastolic function. Reduced LVEF patients are presumed to have LVDD, graded as I, II, or III. Grade I (normal LAP) is E/A ≤ 0.8 (with E ≤ 50 cm/s) or E/A > 0.8 ≤ 2 (or ≤ 0.8 with E > 50 cm/s) with most variables not meeting cut-off values. Grade II (elevated LAP) is diagnosed if E/A > 0.8 ≤ 2 (or ≤ 0.8 with E > 50 cm/s) with ≥50% parameters meeting the cut-off values. Grade III (elevated LAP) is E/A ≥ 2 [24,25].

Various metrics measure LV pressure, including direct LVEDP and indirect LAP via pulmonary capillary wedge pressure (PCWP). LVEDP > 15 mmHg indicates abnormal LV diastolic pressure elevation [26]. Studies reveal that more than 50% of patients with heart failure (HF) have heart failure with preserved ejection fraction (HFpEF), with more than 40% showing isolated LVDD [27]. HFpEF carries high mortality rates that are comparable to patients with heart failure with reduced ejection fraction (HFrEF) [27,28]. As an example, patients with LVDD face higher surgical risks, including prolonged mechanical ventilation, HF exacerbation with increased hospitalization length, and postoperative mortality [29,30,31,32,33,34].

LT candidates may exhibit PH, defined by mean pulmonary artery pressure (mPAP) exceeding 20mmHg [10], further linked to PGD [29]. PH elevates the RV afterload, leading to compensatory RV remodeling and bowing of the interventricular septum. This, in turn, affects LV filling due to ventricular interdependence, potentially causing LVDD [30]. Therefore, LT candidates with PH often display concomitant LVDD [31].

LVDD is highly prevalent in ESLD patients, ranging from 20% to 90% [35,36,37,38,39]. In addition, despite reduced systemic inflammation and RV volume overload subsequently to LT, some recipients may develop LVDD, worsening their vital prognosis [40,41]. 

LVDD is a precursor to HFpEF and remains often ignored, particularly in patients experiencing mixed dyspnea due to concomitant chronic respiratory and cardiac dysfunction [42]. In this regard, the accurate echocardiographic assessment of LVDD is crucial for informed medical decisions, particularly in candidates susceptible to develop PGD. 

## 4. Primary Graft Dysfunction and Left Ventricular Diastolic Dysfunction—Common Mechanisms

Although related to different organs, PGD and LVDD share common mechanisms centered around an imbalanced oxygen demand and supply, promoting IRI [43,44]. Chronic low-grade inflammation, observed in many conditions favoring HFpEF such as metabolic syndrome, COPD, atrial fibrillation, or anemia, constitutes one of the fundamentals of LVDD [45,46]. Cytokine release, prompted by immune reactions, prolonged cell hypoxemia, or the excessive activation of neuroendocrine and autonomic nerve function, may trigger myocardial apoptosis and diminish ventricular compliance [47,48,49]. These processes promote coronary microvascular endothelial inflammation, leading to the generation of reactive oxygen species and reducing nitric oxide bioavailability, as well as cyclic guanosine monophosphate contents and protein kinase G activity in adjacent cardiomyocytes. Ultimately, these factors favor hypertrophy and increase resting tension, reflecting some of the mechanisms observed in PGD [44,45,50,51]. Furthermore, the infiltration of neutrophils into the alveoli is a recognized pathological aspect of IRI in the transplanted lungs, potentially leading to an amplification of cytokine release [45,52]. Finally, proinflammatory cytokines involved in HF pathogenesis include TNF-α, IL-6, IL-8, IL-10, IL-1α, IL-1β, IL-2, TGF-β, and IFN-γ, paralleling those associated with IRI in PGD [53,54]. 

Figure 1 shows these common mechanisms and the complex interplay of the recipient’s LVDD, PH, and RVF in the mechanism of PGD, via an inflammatory cascade, an innate immune system activation, and an increase in pulmonary capillary permeability and pulmonary vascular resistance.

## 5. Primary Graft Dysfunction—The Complex Interplay of Recipient’s Left Ventricular Diastolic Dysfunction, Pulmonary Hypertension, and Right Ventricle Function

Although distinct and considered as a secondary enhancing causative mechanism of PGD, there are interplays of the recipient’s LVDD, PH, and RVF in the complex mechanism of PGD. PH due to left heart disease (group 2 PH) is a common complication of HFrEF and HFpEF [55], of which LVDD may be a precursor [42]. In group 2 PH, there is an increase in the LA pressure that causes a retrograde congestion in the pulmonary circulation, resulting in an increase in pulmonary venous pressure, PCWP, and pulmonary artery pressure (PAP) [56]. The pathophysiology is related not only to an increase in pulmonary arterial pressure but also to an increase in PVR due to adaptations in the vessel walls with hypertrophy and intimal hyperplasia [56]. This increase in PVR is mediated by an increased vascular tone and the remodeling of the pulmonary vasculature, both of which can be triggered by the imbalance between the secretion of vasoactive substances, like endothelin-1 (ET-1), and vasodilated substances, like nitric oxide (NO) and prostacyclin [56]. RV dysfunction in HFpEF is coupled with an elevated afterload, but is not simply due to an afterload -mismatch [57,58,59,60,61,62,63]. Indeed, a partially reversible element of pulmonary vasoconstriction is observed even in patients with an early stage of HFpEF, where a gross volume overload is absent [62]. 

The development of RV dysfunction, from normal to a compensated (hypertrophied) and then a decompensated state, follows the evolution of pulmonary vascular disease [64]. However, after lung implantation in LT, RV afterload is acutely reduced, which increases pulmonary blood flow and the shear stress on the formerly hypoxic pulmonary vascular endothelium. Shear stress leads to capillary leak and impairs graft function [65,66,67,68,69,70,71]. 

## 6. Primary Graft Dysfunction—State of the Art 

Several studies have attempted to elucidate the role of LVDD in PGD following LT, as summarized in Table A1. First, Porteous et al. conducted a retrospective study on patients with interstitial lung disease, COPD, and PAH who underwent bilateral LT between 2004 and 2014 [12]. The diagnosis of PGD was established based on the classical definition, which entails a combination of a PaO_2_/FIO_2_ ratio below or equal to 200 and parenchymal infiltrates on a chest X-ray at 48–72 h after reperfusion [1]. After adjusting for recipient age, BMI, mPAP, and initial respiratory disease, an E/é ratio greater than 8 during pre-transplant check-up was retrospectively associated with a twofold increased PGD risk (OR, 5.29; 95% CI, 1.40–20.01; *p* = 0.01) [12]. Similarly, in a study by Li and colleagues, which included over 300 LT recipients, an elevated LVEDP and PCWP > 15 mmHg were separately associated with a fourfold increased risk of grade 3 PGD, whose reported prevalence was 19% [72]. In contrast, the E/e’ ratio did not significantly differ between LT recipients with and without PGD, possibly due to missing transthoracic echocardiogram (TTE) data. Nevertheless, the authors advocated for a systematic assessment of LVEDP in in LT candidates to more accurately evaluate the risk of PGD, pleading for a multiparametric approach in patients at a higher risk.

Second, Avriel et al. [40] evaluated LVDD using a multiparametric yet partial diagnostic assessment by examining the E/A ratio, prolonged early mitral flow deceleration time, and isovolumic relaxation time in 44 LT candidates. The study demonstrated that the presence of pre-transplant LVDD was associated with increased early postoperative morbidity and decreased 1-year survival [40]. In the sub-group analysis, patients with pre-existing LVDD had a higher risk of extracorporeal life support post-LT (33% vs. 7%, *p* = 0.02) and a trend toward prolonged ventilator-free days [40]. Survival differences post-LT for normal vs. impaired diastolic function emerged in the initial months after LT, underscoring close follow-up with TTE assessment [39,40]. 

Due to continuous improvements in TTE algorithms, PGD evaluations lack current recommendations, creating gaps of evidence on the perioperative LVEDP impact on LT outcomes. Scattered and conflicting data on LVDD pathogenesis in non-PAH LT recipients contribute to inconsistent findings on its predictive value, stemming from different methods analyzing left ventricular filling pressure.

In contrast to Porteous’s study, a retrospective study led by Yadlapati et al. [73] found no correlation between increased E/A, E’/A’ ratios, PCWP, and post-LT events [73]. The limited TTE criteria (2009 guideline) may underestimate LVDD’s prognostic value, affecting medical management and outcomes. Indeed, in addition to the 2009 TTE criteria that included septal e’, lateral e’, and LA volume index [74], in 2016, a new parameter, tricuspid regurgitation velocity, was added for a better quantification of left ventricle filling pressures [24].

To address these limitations, Aggarwal et al. [25] recently explored the impact of LVDD, using the 2016 guidelines criteria in pre-transplant PH recipients [25]. Among 476 primary LT recipients, 205 (mean age of 56.6 +/− 11.9 years; men: 61.5%; mPAP, 30.5 +/− 9.8 mmHg; LVEF < 55% (4.3%)) were analyzed [25]. Pre-transplant, LVDD was present in 93 (45.4%), absent in 16 (7.8%), indeterminate in 89 (43.4%), whereas only 7 (3.4%) patients had missing data for LVDD evaluation. 

Post-transplant, LVDD was present only in 7 LT recipients (3.4%), absent in 164 (80.0%), indeterminate in 15 (7.3%), and 19 (9.3%) had missing data. There was a significant decrease in post-transplant LVDD in those with pre-LT persistent LVDD [25]. Contrastingly, pre-transplant LVDD was not associated with major adverse cardiovascular events (HR 1.08, 95% CI 0.72–1.62; *p* = 0.71) and BOS-free survival (HR 0.67, 74 95% CI 0.39–11.46; *p* = 0.34) at 1-year follow-up [25]. Additionally, a pre-transplant diagnosis of LVDD was not linked to grade 3 PGD [25]. 

In LT candidates with normal LV systolic function and PH, this study found a high prevalence of LVDD, which mostly resolved post-LT regardless of the patients’ characteristics. While these results may seem encouraging regarding LVDD’s impact on LT outcomes, caution is urged due to the study limitations, including its retrospective nature and a significant amount of missing TTE data.

Figure 2 shows the predictive value of different diastolic dysfunction variables in PGD development. Most of the variables were detailed in the studies cited above. The prognostic role of cardiac biomarkers (NT-proBNP or cardiac troponin I) in addition to new echocardiographic parameters designed for a better assessment of RV, right atrial (RA), and LA functions are insufficiently studied in this category of patients; therefore, their role in PGD development in LT candidates is incompletely understood.

## 7. PGD—Controversy and Unresolved Issues 

In their studies, Li and Porteous both identified significant associations between various isolated LVDD parameters and PGD [12,72]. However, the LVDD criteria used by the authors differed between the studies. Unlike Porteous et al., who assessed LVDD by E/e’, Li et al. did not show the additional value of E/e’ in the PGD setting [12,72]. The existence of conflicting data underscores the significance of a multiparametric evaluation of LVDD. A common factor of the earlier reports on PGD risk evaluation was the presence of PH. PH increases RV afterload, causing compensatory RV remodeling, further inducing LV dysfunction, translated into high filling pressures in the context of preserved systolic function [40]. Consequently, the presence of LVDD (secondary to PH) might be a cofounding factor in this setting as some of the cardiac morphological and functional abnormalities secondary to PH tend to be resolved following LT [75,76]. Xie et al. found that the LV E/A ratio and atrial filling fraction persist beyond two months post-LT but returned to normal at one-year follow-up [41], suggesting the reversibility of LV perioperative parameters. Considering that PGD occurs early after LT, the prompt detection of de novo abnormal LV diastolic function post-LT is crucial, impacting clinical decisions amid advancements in medical management across diverse scenarios. The use of novel medications, like sodium glucose transporter 2 inhibitors (SGLT2-i), in individuals with kidney or heart transplants demonstrated both safety and beneficial outcomes [77,78]. In left ventricular assist device patients, a notable reduction of 5.6 mmHg in sPAP was observed following SGLT2-i initiation [79]. While more extensive studies involving solid organ transplant patients are currently needed, this limited hypothesis-generating report suggest that this class of medication, widely employed in various LVDD conditions, is safe and effective, improving patients’ survival and quality of life, while reducing healthcare costs [80]. This further highlights the necessity of active LVDD screening in the LT population in order to implement PGD-preventive measures. Promoting research programs focused on better understanding the heterogeneity of PGD mechanisms, in addition to developing new treatment algorithms, is mandatory for patients’ wellbeing. Advancements in cardiovascular imaging, such as 4D evaluation and deformation imaging employing strain and myocardial work analysis, hold the potential to enhance our comprehension of the mechanisms underlying PGD. These technologies aid in identifying subclinical dysfunction, providing a decisive impact on the future medical care of patients.

## 8. Conclusions

Primary graft dysfunction is the predominant cause of lung transplantation’s early mortality, yet the mechanisms of its onset remain inadequately understood. Left ventricular diastolic dysfunction, indicated by left ventricle end-diastolic pressure, may contribute to an elevated risk of primary graft dysfunction. However, discrepancies in the existing data underscore the importance of adopting a multiparametric approach, as recommended by the current clinical practice guidelines. Future research with standardized protocols is essential for a more precise understanding of left ventricle end-diastolic pressure’s prognostic value in post-lung transplantation outcomes and for refining clinical management strategies.

## Figures and Tables

**Figure 1 diagnostics-14-01340-f001:**
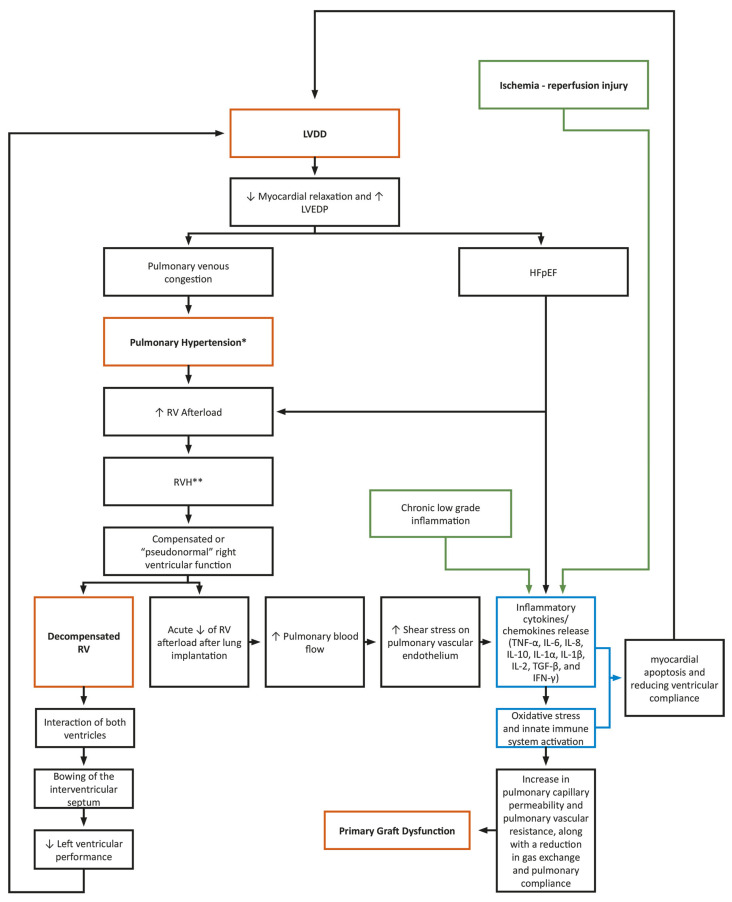
The common mechanisms of PGD and LVDD and the interplay of the recipient’s LVDD, PH and RVF in the mechanism of PGD. HFpEF: Heart Failure with preserved Ejection Fraction; LVDD: Left Ventricular Diastolic Dysfunction; LVEDP: Left Ventricular End-Diastolic Pressure; RV: Right Ventricular; RVH: Right Ventricular Hypertrophy; * Mean pulmonary artery pressure > 20 mmHg, measured by right heart catheterization; ** Defined as a wall thickness greater than 5 mm.

**Figure 2 diagnostics-14-01340-f002:**
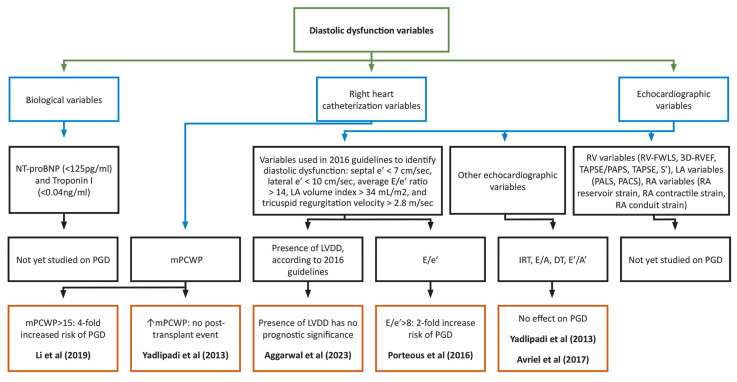
Predictive value of different diastolic dysfunction variables on PGD. Aggarwal et al.: [25]; Avriel et al.: [40]; DT: Deceleration Time; IRT: Isovolumic Relaxation Time; LA: Left Atrial; LVDD: Left Ventricular Diastolic Dysfunction; Li et al.: [72]; mPCWP: Mean Pulmonary Capillary Wedge Pressure; PACS: Peak Atrial Contraction Strain; PALS: Peak Atrial Longitudinal Strain; PAPs: Pulmonary Arterial Systolic pressure; PGD: Primary Graft Dysfunction; Porteous et al.: [12]; RA: Right Atrial; RV: Right Ventricle; 3D-RVEF: Three-Dimensional Echography Right Ventricular Ejection Fraction; RV FWLS: Right Ventricular Free Wall Longitudinal Strain; TAPSE: Tricuspid Annular Plane Systolic Excursion; Yadlipadi et al.: [73].

**Table 1 diagnostics-14-01340-t001:** Risk factors for the development of primary graft dysfunction (PGD) following lung transplantation.

Category	Risk Factors [14,18,19]
Donor	- Age (>45 years or <21 years); - American and African race; - Female sex; - Hemodynamic instability after brain death; - Prolonged mechanical ventilation; - Smoking history.
Recipient’s variables	- Body mass index > 25; - Diagnosis: IPF, PAH, PH secondary to parenchymal lung disease, sarcoidosis; - Elevated pulmonary arterial pressure at time of surgery; - Female sex.
Operative variables	- Single lung transplantation; - Prolonged ischemic time; - Use of cardiopulmonary bypass; - Packed red blood transfusion > 1 L; - High reperfusion FiO_2_ > 0.4.

IPF: Idiopathic Pulmonary Fibrosis; PAH: Pulmonary Arterial Hypertension; PH: Pulmonary Hypertension.

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
