# Peer review of "Impact of Pre-Transplant Left Ventricular Diastolic Pressure on Primary Graft Dysfunction after Lung Transplantation: A Narrative Review"

_diagnostics, 2024, doi:10.3390/diagnostics14131340_

Round 1
Reviewer 1 Report
Comments and Suggestions for Authors
Henry et al reported their work named " Impact of Pre-Transplant Left Ventricular Diastolic Pressure on Primary Graft Dysfunction after Lung Transplantation: A Narrative Review" and concluded "In this review, we aim at summarizing PGD mechanisms, risk factors and diagnostic criteria, with further focus on the interplay between LVDD and PGD development. Finally, we explore the predictive value of several diastolic dysfunction diagnostic parameters to predict PGD occurrence and severity.". I have the following minor comments:
- Please don't write Authors' ORCID in this way as the journal will ask you about them later.
- Please avoid abbreviations in the conclusion and sections title eg PH and RVF
- Please spell out abbreviations at their first time eg line 181: HFrEF and HFpEF
- Section 4 "4. PGD and LVDD – Common Mechanisms"; please cite your figure 1 here.
- Please try to add colors to your figures.
- Table A1: Please transfer this table as a main table or transform it into a colored figure. Please specify if there is a certain age cutoff or bimodal distribution
- Table A2: Please make the page in portrait (horizontal) orientation. Please specify the meaning of "LVDD indet." in Aggarwal 2023 study.
Comments on the Quality of English LanguageMinor edits are essential.
Author Response
"Please see the attachment."

Reviewer 2 Report
Comments and Suggestions for Authors
This manuscript entitled “the Impact of Pre-Transplant Left Ventricular Diastolic Pressure on Primary Graft Dysfunction after Lung Transplantation” summarized the mechanisms of Primary Graft Dysfunction, risk factors and diagnostic criteria, further focusing on the interplay between LVDD and PGD development. They also explored the predictive value of several diastolic dysfunction diagnostic parameters to predict occurrence and severity. The organization of this review is good, and easy to understand. I have several concerns as followed:
1. The head of all tables have to be revised. Do not use “This table summarized...”. Directly describe the main points of the tables as the one of your figures.
2. The authors can add a figure to summarize the major mechanisms for the section “PGD and LVDD – Common Mechanisms”.
3. Some grammar errors should be corrected.
Comments on the Quality of English LanguageNA
Author Response
"Please see the attachment."
